# Improving Performance and Breakdown Voltage in Normally-Off GaN Recessed Gate MIS-HEMTs Using Atomic Layer Etching and Gate Field Plate for High-Power Device Applications

**DOI:** 10.3390/mi14081582

**Published:** 2023-08-11

**Authors:** An-Chen Liu, Po-Tsung Tu, Hsin-Chu Chen, Yung-Yu Lai, Po-Chun Yeh, Hao-Chung Kuo

**Affiliations:** 1Department of Photonics, Institute of Electro-Optical Engineering, National Yang Ming Chiao Tung University, Hsinchu 30010, Taiwan; arsen.liou@gmail.com (A.-C.L.); itria30378@itri.org.tw (P.-T.T.); 2Electronic and Optoelectronic System Research Laboratories, Industrial Technology Research Institute, Zhudong 310401, Taiwan; a50164@itri.org.tw; 3Institute of Advanced Semiconductor Packaging and Testing, National Sun Yat-sen University, Kaohsiung 804201, Taiwan; 4Research Center for Applied Sciences, Academia Sinica, 128 Sec. 2, Academia Rd., Nankang, Taipei 115, Taiwan; loveriver031@gmail.com; 5Semiconductor Research Center, Hon Hai Research Institute, Taipei 114699, Taiwan

**Keywords:** atomic layer etching (ALE), MIS-HEMT, gate surface roughness

## Abstract

A typical method for normally-off operation, the metal–insulator–semiconductor-high electron mobility transistor (MIS-HEMT) has been investigated. Among various approaches, gate recessed MIS-HEMT have demonstrated a high gate voltage sweep and low leakage current characteristics. Despite their high performance, obtaining low-damage techniques in gate recess processing has so far proven too challenging. In this letter, we demonstrate a high current density and high breakdown down voltage of a MIS-HEMT with a recessed gate by the low damage gate recessed etching of atomic layer etching (ALE) technology. After the remaining 3.7 nm of the AlGaN recessed gate was formed, the surface roughness (Ra of 0.40 nm) was almost the same as the surface without ALE (no etching) as measured by atomic force microscopy (AFM). Furthermore, the devices demonstrate state-of-the-art characteristics with a competitive maximum drain current of 608 mA/mm at a V_G_ of 6 V and a threshold voltage of +2.0 V. The devices also show an on/off current ratio of 10^9^ and an off-state hard breakdown voltage of 1190 V. The low damage of ALE technology was introduced into the MIS-HEMT with the recessed gate, which effectively reduced trapping states at the interface to obtain the low on-resistance (R_on_) of 6.8 Ω·mm and high breakdown voltage performance.

## 1. Introduction

Owing to its excellent electrical properties, gallium nitride (GaN) power devices have received enormous attention for next-generation switching power devices [1]. Currently, the dominant structure of GaN devices that is commercially available is based on a lateral structure design, which is grown on large low-cost silicon (Si) wafers. GaN on Si can be fabricated using the existing fully mature 6 or 8 in silicon fabrication facilities, which further offers the advantages of the cost competitiveness of GaN on Si power technology. In addition, the rapid advance in device design, epitaxial growth, gate-driving techniques, processing, and packaging technology has made the possibility of the commercialization of GaN on Si power devices, which are commonly implemented in compact and efficient power converters [2].

The GaN devices are divided into two types according to the mode of operation, enhancement mode (E-mode), or normally-off, and depletion mode (D-mode), or normally- on. For power applications, E-mode devices are preferred as this provides less power loss during switching, safer operation, and less-complex gate driver circuitry [3]. There are different varieties of approaches for designing E-mode GaN devices; among them, the most popular ones are by using the recessed gate GaN MIS-HEMT [4,5], the p-GaN HEMT [6,7,8], fluoride treatment [9], and the cascode configuration circuit [10,11]. In fluoride treatment, a negative charge is introduced under the gate electrode to adjust the threshold voltage (*V_TH_*) of the device to make it a normally-off device. However, there is an issue regarding the stability of *V_TH_* during high-temperature annealing [12]. In the p-GaN HEMT, a p-GaN cap is introduced below the Schottky gate, which lifts the conduction band and, thus, achieves a normally-off device. The device exhibits lower on-resistance and also has high mobility. However, the device suffers from the incapability of achieving high *V_TH_* and also has various issues regarding gate reliability [13]. Among the above-mentioned methods, the implementation of the recessed gate GaN MIS-HEMT is more widely considered as it provides a larger gate swing, higher *V_TH_*, and lower gate leakage. Methods available for performing gate recess are the dry etch process [14] and the chlorine-based dry etch process [15].

A procedure that is commonly used for performing the recessed gate structure is by using dry etching techniques, namely inductively coupled plasma-reactive ion etching (ICP-RIE) [16,17,18,19,20]. However, the UV photons present in the plasma discharged from the process cause serious damage to the surface of the semiconductor [21,22]. The damage on the surface causes an increase in leakage current, *V_TH_* instability, and current collapse, hence degrading the performance of the device [23,24,25,26]. Furthermore, in the conventional method, the suppression of the remaining AlGaN thickness variation is difficult to perform at the nanometric scale level, as the processing time controls the amount of etching [27]. An alternative etching process that provides high-quality interface engineering after etching is atomic layer etching (ALE). The ALE process is a type of dry etching technique that uses a series of self-limiting surface reactions to remove material layer by layer with high precision and control. In the case of GaN HEMT fabrication, the ALE process is used to create a recessed gate structure in the AlGaN/GaN heterostructure and has low etching damage [28,29,30,31,32,33,34]. In this study, a high breakdown voltage of 1190 V of the normally-off AlGaN/GaN MIS-HEMT with a recessed gate was successfully demonstrated. The ALE process was used instead of using ICP-RIE in order to reduce the roughness of the gate recess surface and further improve its trapping states at the interface.

The normally-off AlGaN/GaN MIS-HEMT with a recessed gate was verified using the ALE process with the Cl_2_/BCl_3_ reaction gas to carefully control the gate recess process and minimize surface damage. Furthermore, atomic layer deposition (ALD) was used to create high-gate dielectric Al_2_O_3_ for reduced leakage current, lower interface trap density, and improved surface passivation. At an off-state current density of 1 mA/mm, an extremely low gate leakage of 10^−7^ mA/mm and a dramatically high breakdown voltage of 1190 V were achieved.

## 2. Devices’ Fabrication

The fabricated normally-off AlGaN/GaN MIS-HEMT with a recessed gate structure was grown on a 6 in commercial Si substrate by MOCVD, which consisted of the 1 nm GaN cap layer, a 25 nm Al_0.25_Ga_0.75_N barrier layer, a 1 nm AlN interlayer, a 300 nm undoped GaN channel layer, and a 4 µm AlGaN buffer layer [35,36], as shown in Figure 1a. The normally-off MIS-HEMT with a recessed gate process started with mesa isolation, which was performed by the ICP-RIE to etch a specific region. Subsequently, the ohmic contact (S/D metal) based on Ti/Al/Ni/Au (25/125/45/75 nm) was formed by rapid thermal annealing (RTA) at 825 °C for 30 s in nitrogen (N_2_) atmosphere to complete the source–drain ohmic connections. The ohmic contact resistance was 0.6 Ω mm. Then, a significant recessed gate structure was formed by the ALE technology. For a low-damage surface in the AlGaN barrier by the ALE process, a recessed gate pattern was fabricated via Cl_2_/BCl_3_ hybrid etching.

In this case, the GaN cap and AlGaN barrier were recessed by multiple cycles of oxidation and etching using inductively linked plasma, with each cycle consisting of generating a thin oxide layer on top and etching away using BCl_3_ etching [37]. ALE employing two quasi-self-limiting steps (O_2_ plasma modification and BCl_3_ plasma removal) was used to fabricate the AlGaN/GaN HEMT. This technique allows for the precise control of recessing and the suppression of leakage current, making it a promising alternative to the continuous dry etching method [38,39,40].

A wet cleaning process of HCl:DI = 1:10, BOE:DI = 1:10, and remote O_2_ plasma pretreatment was applied to remove the metal impurities, carbides, and native oxide, followed by the ALD of the 11.45 nm Al_2_O_3_ high-dielectric layer synthesized using trimethylaluminum and H_2_O as precursors at 350 °C. A gate metal with Ni/Au (50 nm/200 nm) was deposited by an E-Gun evaporator on the Al_2_O_3_ high-dielectric layer, followed by the first layer passivation of SiNx with 200 nm deposited by plasma enhanced chemical vapor deposition (PECVD). To suppress the high-electric-field effect between the gate-to-source/drain, a gate field plate with a width of 11 µm with field plate distance extending from the edge of the gate to the drain (L_GFP_) of 3.5 µm was fabricated by lithograph patterning and a Ni/Au (50 nm/200 nm) metal stack deposited by the E-Gun. Then, the second layer passivation of SiNx with a 200 nm thickness was fabricated. For the final pad-contacting hole opening in the gate, source, and drain region defined by the lithograph process, the last pad metal stack of Ni/Au (50 nm/ 200 nm) was deposited by the E-Gun. The specific characteristic values (feature length) of the MIS-HEMT with the recessed gate were the gate length (L_G_) of 3 µm, the gate width (W_G_) of 100 µm, the gate-to-source length (L_GS_) of 5 µm, the gate-to-drain length (L_GD_) of 10 µm, as shown in Figure 1a,b. Figure 1c presents the SEM image of the single finger in the recessed gate region, clearly showing the gate metal, gate field plate, and SiNx passivation.

The gate metal Ni/Au (50 nm/200 nm) structure deposited by the E-Gun is a crucial component of the AlGaN/GaN HEMT, and its design can significantly impact the device’s performance. Typically, the gate metal structure consists of a thin layer of refractory metal, such as gold, deposited on the semiconductor surface to create the gate electrode. It affects critical parameters such as gate-to-channel capacitance, gate resistance, and gate-to-drain/source capacitance, which can significantly impact the device’s performance. In this study, we found that using Ni/Au as the gate metal structure can lead to a low gate resistance and a low gate-to-channel capacitance, ultimately improving the device’s transconductance.

The gate field plate structure is a crucial element in the AlGaN/GaN HEMT. Typically, it consists of a thin layer of metal or doped semiconductor that connects to the gate contact and extends beyond the source and drain edge. This structure effectively increases the gate-to-source/drain distance and reduces the electric field at the gate edge, leading to an improvement in the device’s breakdown voltage and a reduction in the gate leakage current. The gate field plate design significantly impacts the device’s performance, affecting its breakdown voltage, gate leakage current, and parasitic capacitance. In this paper, Ni/Au was utilized as the gate field plate material, and a large-area structure with a width of 11 µm was designed to effectively reduce the gate leakage current and improve the device reliability. However, this may increase parasitic capacitance, thereby degrading the high-frequency performance of the device. The selection of the gate metal and gate field plate structures should be carefully considered based on the specific application requirements to optimize the device’s performance and reliability. Using Ni/Au as the gate metal in HEMT provides good electron transport performance, contributing to the high-frequency characteristics and fast switching speed of the device. The Ni/Au layer has good adhesion with HEMT materials, typically being semiconductor materials. This ensures a strong contact between the metal gate and the semiconductor, thereby improving the device’s performance and reliability. Au is highly resistant to oxidation, meaning the metal gate is less prone to oxidation during long-term usage. This contributes to the device’s longevity and stability. The common choices of the metal for the gate field plate are Ni/Au [41] or Ti/Au [42], both of which have excellent conductivity and adhesion.

Compared to other metals, the main differences of Ni/Au lie in their conductivity and adhesion properties. While other metals may have good conductivity, they may not adhere as well to semiconductors. Poor adhesion could lead to unstable performance or failure of the device. Additionally, some metals may be more susceptible to oxidation, reducing the device’s reliability. Therefore, considering factors such as conductivity, adhesion, and oxidation resistance, Ni/Au is widely used in HEMT devices to ensure high performance and reliability. The gate field plate width designed at 11 µm effectively disperses the electric field, improving the breakdown voltage. As indicated in the reference, the lower the proportion of the L_GFP_ length to the L_GD_ length, the higher the maximum transconductance (Gm) value and the gain will be, which is mainly due to the reduction in the access resistance caused by the increase in available carriers in the channel. According to A. Mohanbabu et al. [43], when the L_GFP_/L_GD_ ratio is around 30%, it can achieve better Gm and breakdown voltage [43,44,45].

The *V_TH_* for the AlGaN/GaN MIS-HEMT with a recessed gate is expressed as Equation (1) [46,47].
(1)$$\;\;\;V_{TH}=\frac{1}{e}\left(\varphi_{B}\left(x\right)+E_{F}\left(x\right)-∆E_{C1}\left(x\right)-∆E_{C2}\left(x\right)\right)-\sigma \left(x\right)\left(\frac{t_{die}}{\varepsilon_{0}\varepsilon_{Al2O3}}+\frac{t_{RB}}{\varepsilon_{0}\varepsilon_{AlGaN}}\right)$$
In the equation, $\frac{t_{RB}}{\varepsilon_{0}\varepsilon_{AlGaN}}$ represents the layer thickness ($t_{RB}$) and dielectric constant (ε) of the AlGaN layer, respectively. $\left(\frac{t_{die}}{\varepsilon_{0}\varepsilon_{Al2O3}}\right)$ represents the layer thickness ($t_{die}$) and dielectric constant (ε) of the Al_2_O_3_ layer, respectively. $\sigma \left(x\right)$ is the polarization sheet charge amount. $E_{F}\left(x\right)\;$ is the difference between the Fermi level and the GaN conduction band edge level. $∆E_{C1}\left(x\right)\;$ and $∆E_{C2}\left(x\right)\;$ denote the conduction band discontinuity between Al_2_O_3_ and AlGaN and between AlGaN and GaN, respectively. $\varphi_{B}\left(x\right)$ is the Schottky barrier height.

The movement of *V_TH_* in the positive direction is strongly correlated with the AlGaN barrier thickness, as a decrease in thickness leads to a reduction in both the 2DEG concentration and electron mobility, ultimately resulting in a positive shift in *V_TH_*. However, a high *V_TH_* can negatively affect the drain current (I_D_) due to the reduced 2DEG concentration. Therefore, the precise control of the AlGaN layer etch depth is critical to optimize both *V_TH_* and I_D_ current. To avoid potential dielectric (Al_2_O_3_) side effects, we investigated three different depths of 5 nm and 3 nm and over-etching of the AlGaN gate recess in Schottky HEMT devices, and the reference device was without the gate recessed etching. By examining the correlation between the threshold voltage shift and AlGaN depths, we found that AlGaN with a 3 nm remaining thickness of the AlGaN layer showed the realization of normally-off characteristics in the HEMT devices. Although the over-etching of the AlGaN gate recess also had normally-off characteristics, the lowest drain current was 6 mA/mm. AlGaN with 3 nm remaining showed a threshold voltage of +0.5 V and a drain current of 440 mA/mm at a gate voltage of 4 V, and the reference device without gate recessed etching showed a threshold voltage of −4.5 V and a drain current of 650 mA/mm. The 3 nm remaining thickness of the AlGaN was inspired by a prior study, which yielded promising outcomes with this particular approach in the recessed gate MIS-HEMTs. The maximum drain current density of a typical D-mode HEMT can reach 900 mA/mm. With the decrease in the AlGaN thickness, the 2DEG concentration and electron mobility will also decrease. Theoretically, the critical value for generating a 2DEG is about 3 nm [48]. When AlGaN is 0 nm, the current reaches its lowest, indicating very limited electrons in the channel. When there is little 2DEG in the channel under the gate, the scattering of electrons increases, and the electron mobility is very low. The *V_TH_* in this paper is defined as the gate bias when the drain current reaches 1 mA/mm. The *V_TH_* of a typical D-mode HEMT is about −4 V. When the thickness of AlGaN decreases, the *V_TH_* of the MIS-HEMT also rises. This indicates that *V_TH_* can be determined by controlling the etching depth. Secondly, when the dielectric layer of Al_2_O_3_ is added, the capacitance between the dielectric layer and AlGaN increases, further reducing the 2DEG concentration and electron mobility. Therefore, we can achieve the best device characteristics by adjusting the gate etching depth and the thickness of the dielectric layer [46].

The depth of the gate recess affects the carrier transport characteristics of the device. Typically, a deeper gate recess can increase the width of the carrier channel, reducing resistance, enhancing carrier mobility, and improving the current-carrying capacity, and the depth of the gate recess affects the *V_TH_* of the device. Generally, a deeper gate recess leads to a higher threshold voltage, making the device more conducive to switching operations. Therefore, the depth of the gate recess requires a trade-off [46]. The ALE process is a promising technology for achieving nanoscale depth control and low surface damage in the fabrication of recessed gate MIS-HEMT devices, providing an excellent platform for further improving device performance.

Here, we discuss the characterization of a recessed gate structure in AlGaN layers using different etching techniques. We also highlight the advantages of ALE over traditional plasma etching methods in terms of surface roughness and damage. Transmission electron microscopy (TEM) analysis was conducted on the recessed gate structure region, as shown in Figure 2a. The TEM profile revealed the thicknesses of the Al_2_O_3_ and AlGaN layers to be 11.45 nm and 3.70 nm, respectively. This analysis provides important information about the dimensions of the layers in the structure. Conventionally, GaN-based materials are etched using the inductively coupled plasma-reactive ion etching (ICP-RIE) system, which is commonly employed for plasma etching processes such as gate recesses and ohmic recesses. However, one drawback of this technique is the potential lattice damage caused by the generation of free radicals, ions, and ultraviolet light during plasma discharge, as mentioned in [41].

On the other hand, alternative etching processes such as ALE have been found to exhibit relatively low surface roughness compared to digital etching and continuous etching processes using Cl_2_/BCl_3_. This means that ALE can potentially minimize the surface damage during the etching process. To demonstrate the low surface damage achieved through the ALE process on the AlGaN layer, the surface roughness in the recessed gate region was measured using AFM. In Figure 2b, the arithmetic average roughness (Ra) is reported as 0.40 nm. This value indicates that the recess etching using ALE resulted in remarkably low surface damage, which was very similar to the surface without any etching. This finding suggests that ALE can be a promising technique for achieving precise etching while minimizing surface roughness and damage in AlGaN layers. Overall, here, an overview of the TEM analysis of a recessed gate structure is given, comparing traditional plasma etching techniques with ALE in terms of lattice damage and surface roughness and providing evidence for the low surface damage achieved through the ALE process using AFM measurements.

## 3. Results and Discussion

In order to know the transfer and gate leakage characteristics of the normally-off AlGaN/GaN MIS-HEMT with the recessed gate, the device design with L_G_/L_GS_/L_GD_/W_G_ = 3/5/10/100 µm of the drain current–gate bias (I_D_–V_G_) measurement was performed. The fabricated device was a single-finger device, as shown in Figure 1c, with transfer characteristics in a linear scale, as shown in Figure 3a. According to the gate bias of the maximum transconductance curve, *V_TH_* was determined to be +2.0 V by linear extrapolation or +0.7 V at a drain current extracted at 1 µA/mm. The maximum drain current (*I*_D, max_) of 608 mA/mm at V_G_ of 6 V and G_m_ of 170 mS/mm at V_G_ of 2 V were demonstrated, respectively. Figure 3b shows the semi-log scale of the I_D_–V_G_ curve; the drain bias (V_D_) was operated at 10 V, and the gate voltage (V_G_) was swept from −2 V to 8 V. A subthreshold swing (SS) was found in 116 mV/dec. Even with a significant forward bias of 10 V, the I_D_-to-I_G_ ratio was more than 10^9^, indicating that the high dielectric leakage current was negligible. This was due to the low-surface-damage etching on the AlGaN layer by the ALE process. A *V_TH_* hysteresis of 0.6 V was found. The *V_TH_* hysteresis phenomenon was produced by electrons caught in the AlGaN/Al_2_O_3_ dielectric interface states or traps that remained in the Al_2_O_3_ dielectric bulk layer. Compared to other devices, the hysteresis of this device was relatively low [49,50].

Figure 4a shows the DC I_D_–V_D_ output characteristics of a normally-off AlGaN/GaN MIS-HEMT with the recessed gate device with the feature design with L_G_/L_GS_/L_GD_/W_G_ of 3/5/10/100 µm. V_G_ was applied from −1 V to 4 V, and the maximum output drain current density was 556 mA/mm at a V_D_ of 10 V. Furthermore, the R_on_ of the recessed gate device was 6.8 Ω·mm (corresponding specifically to on-resistance, R_ON, SP_ was 1.27 mΩ·mm^2^). The three-terminal off-state breakdown voltage (BV) characteristics of the normally-off MIS-HEMT with the recessed gate device is shown in Figure 4b. The recessed gate device feature design with L_G_/L_GS_/L_GD_/W_G_ of 3/5/10/100 µm and V_G_ of −10 V was used as the BV measured condition. The BV was defined as the point at which the gate and drain leakage current reached 1 µA/mm and 1 mA/mm. The device’s hard breakdown was dominated by the gate-to-drain leakage current (V_D_) = 720 V at 1 µA/mm. One Al_2_O_3_ gate dielectric design layer and the low-surface-damage recessed gate by the ALE process improved the gate leakage current. Compared to the study of Hsieh et al., using the Al_2_O_3_/AlN stack insulator reduced the interface trapping density between the Al_2_O_3_/GaN interface for the gate recessed GaN MIS-HEMT made by low-power-plasma etching, for which the hard breakdown occurred at a gate-to-drain voltage (V_D_) = 620 V at 2 µA/mm [20]. The high gate-to-drain leakage current mechanism is due either to surface conduction or to the tunneling of electrons through the AlGaN barrier [51]. According to the gate-to-drain breakdown voltage results, there was a three-step breakdown behavior. The slight decrease of the I_G_ current before a V_D_ of 200 V in the initial stages indicated trapping, and the beginning of the increase of the I_G_ current from 200 V to 1100 V after that can be attributed to the stress-induced leakage current (SILC) [52]. When applied to the device, electrical stress can cause structural changes in surface conduction or the oxide layer, creating traps that can capture and release charge carriers. As a result, these traps can influence the current flow through the device, leading to leakage currents. Dielectric hard breakdown, the catastrophic breakdown, was observed to take place after 1100 V and was characterized by a very sharp vertical jump in the gate current of each device [53]. Therefore, a BV was exhibited of 720 V at 1 µA/mm and another BV was exhibited of 1190 V at 1 mA/mm. Owing to the low-damage surface of the recessed gate, which can be well covered by the high-quality dielectric Al_2_O_3_, the high critical breakdown field strength between the gate and the drain region can be effectively suppressed by the gate field plate design. A low surface gate leakage current was demonstrated to achieve a high breakdown voltage performance of 1190 V. Additionally, the gate field plate design also helped achieve a higher BV performance for the recessed gate devices.

To further compare the trade-off between *V_TH_* and ID_max_, Table 1 highlights the performance of the proposed MIS-HEMT with the recessed gate device and the state-of-the-art normally-off GaN devices. Compared to the device discussed in [54], our study exhibited a slightly higher BV of 720 V at 1 µA/mm. However, their *I*_D, max_ exceeded this work by 200 mA/mm. This was due to the double-channel heterostructure design, which offered high 2DEG mobility by the lower channel heterostructure. The upper channel heterostructure achieved a fully recessed gate by completely etching away the AlGaN layer. This fully recessed gate structure aimed to obtain a normally-off operation with a positive *V_TH_* requirement. Typically, when the gate recess is fully etched away, *V_TH_* tends to be large, leading to a lower *I*_D, max_. Therefore, the design of a double-channel heterostructure necessitates carefully considering the complex epitaxy process to attain high positive *V_TH_* and *I*_D, max_. In contrast, this work with a standard one-channel heterostructure and by ALE etching retains a remaining 3.7 nm of AlGaN after the gate recess, providing a simple process and controllable adjustment of *V_TH_* and *I*_D, max_.

The MIS-HEMT with the recessed gate formation using the novel low-damage ALE technology showed competitive performance, which featured both a high current density of 608 mA/mm and a controllable threshold voltage of +2.0 V. Another important factor for power GaN devices is the BV and *R*_ON,SP_. The ALE-fabricated recessed gate devices demonstrated a very competitive BV of 1190 V at 1mA/mm and *R*_ON,SP_ of 1.27 mΩ·cm^2^ when compared to some state-of-the-art normally-off GaN devices. These results suggest that the normally-off device with the ALE recessed gate has significant potential for high-power device applications, such as in power converters and inverters.

## 4. Conclusions

Our study successfully showcased the exceptional performance of a normally-off GaN MIS-HEMT device with a recessed gate. Using the novel ALE technology allowed us to precisely control the recessed gate with normally-off characteristics with a remaining thickness of 3.7 nm, providing distinct advantages in nanoscale control and minimal surface damage. The device exhibited high-performance characteristics, including a high maximum drain current (*I*_D, max_) of 608 mA/mm, a controllable threshold voltage (*V_TH_*) of +2.0 V, and an outstanding on/off current ratio of 10^9^. Furthermore, the device demonstrated a remarkable off-state hard breakdown voltage (*BV*) of 1190 V at 1 mA/mm in the gate-to-drain of the I_G_ current. The superior BV performance can be attributed to the low surface roughness (Ra) of 0.40 nm in the recessed gate region, ensuring excellent coverage of the high-quality dielectric Al_2_O_3_. Additionally, incorporating a gate field plate with a width of 11 µm in the design effectively suppressed the high critical breakdown field strength between the gate and drain regions, further enhancing the device performance. Based on these studies, we believe that normally-off GaN MIS-HEMTs with recessed gates, enabled by the innovative ALE technology, hold significant potential for applications in large-area high-power devices. These devices can achieve remarkable BV and *I*_D, max_ values and maintain a controllable V_TH_ and a high on/off current ratio, making them up-and-coming candidates for next-generation power electronics applications.

## Figures and Tables

**Figure 1 micromachines-14-01582-f001:**
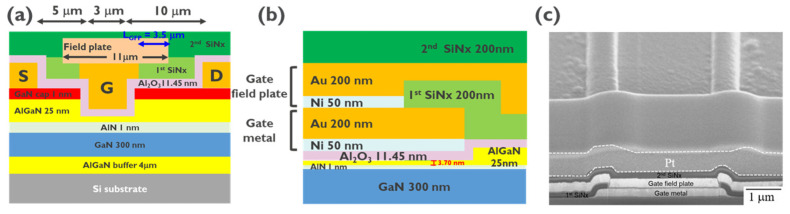
(**a**) Cross-section schematic view of the AlGaN/GaN MIS-HEMT with a recessed gate. (**b**) MIS gate stack structure. (**c**) SEM image of the MIS-HEMT with a recessed gate.

**Figure 2 micromachines-14-01582-f002:**
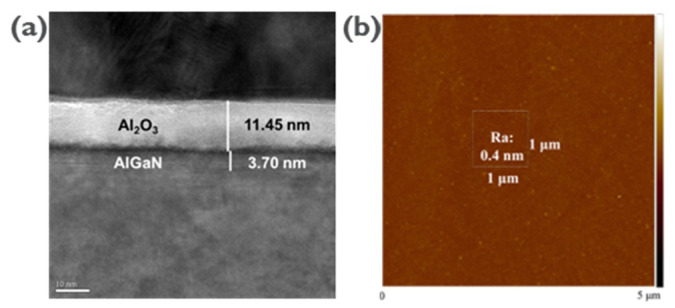
(**a**) The recessed gate structure from the TEM image and (**b**) the surface morphology of the recessed gate region; Ra of 0.4 nm by AFM inspection.

**Figure 3 micromachines-14-01582-f003:**
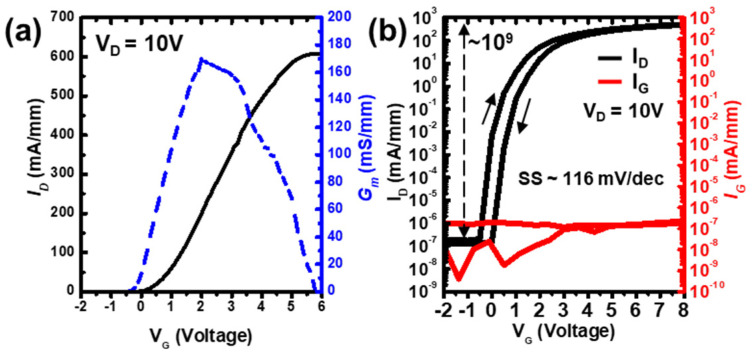
Transfer characteristic of normally-off AlGaN/GaN MIS-HEMT with the recessed gate: (**a**) I_D_–V_G_ characteristic (black solid line) and G_m_ characteristic (blue dotted line) in a linear scale; (**b**) I_G_–V_G_ sweeps characteristic (red solid line) and I_D_–V_G_ sweeps characteristic (black solid line) in a semi-log scale.

**Figure 4 micromachines-14-01582-f004:**
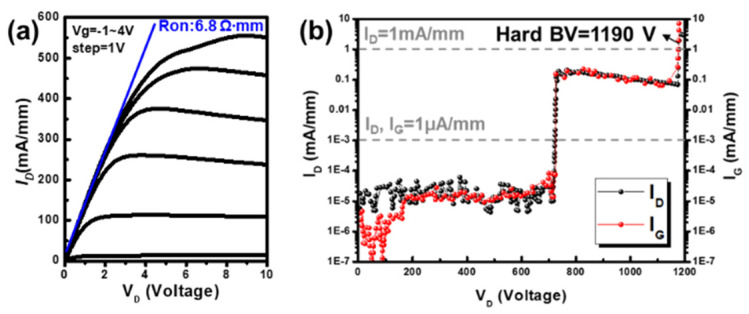
(**a**) DC I_D_–V_D_ output characteristics of the normally-off AlGaN/GaN MIS-HEMT with the recessed gate. (**b**) Off-state breakdown characteristics of the recessed gate device with a gate–drain length of 10 µm measured at V_G_ of −10 V.

**Table 1 micromachines-14-01582-t001:** Comparison of state-of-the-art normally-off GaN devices [54,55,56,57,58,59,60,61,62].

Ref.	*V_TH_* (V)	*I*_D, max_ (mA/mm)	*R*_ON,SP_ (mΩ·cm^2^)	*BV* (V)
This work	2.0	608	1.27	720 at 1 µA/mm 1190 at 1 mA/mm
[54]	0.5	836	1.48	705 at 1 µA/mm
[55]	1.35	500	1.56	1400 at 5 µA/mm
[56]	0.4	356	2.79	880 at 5 µA/mm
[57]	1.5	110	1.5	135 at 1 mA/mm
[58]	3.2	630	3.3	696 at 34 mA/mm
[59]	3.0	363	0.87	650 at 1 µA/mm
[60]	1.2	825	0.63	810 at 1 mA/mm
[61]	0.8	312	2.73	852 at 1 µA/mm
[62]	2.6	823	1.76	710 at 10 µA/mm

## Data Availability

Not applicable.

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
