# Peer review of "Improving Performance and Breakdown Voltage in Normally-Off GaN Recessed Gate MIS-HEMTs Using Atomic Layer Etching and Gate Field Plate for High-Power Device Applications"

_micromachines, 2023, doi:10.3390/mi14081582_

Round 1

Reviewer 1 Report

In this paper, the authors demonstrate a high current density and high breakdown voltage of MIS-HEMT with recessed gate by the low damage gate recessed etching of atomic layer etching (ALE) technology. Though this work could be interesting, the reviewer thinks that it requires major revisions before it can be considered for publication. Detailed comments are listed as follows:

1.      We noticed that many illustrations in this paper were repeated and the statements should be checked carefully. For example, in Page 2 Line 74, the authors stated that “…and keeps low etching damage and keep low etching damage.” In Page 3 Line 122 and 123, the authors stated that “gate width (Wg) of 100 µm, … and the gate width (WG)of 100 µm.” In Page 3 Lin 140 and 147, the authors stated that “The gate field plate… affecting its breakdown voltage, gate leakage current, …the gate field plate structure affects the device’s breakdown voltage and gate leakage current.”

2.      In Page 2, line 91, the authors stated that the GaN buffer layer was 4 µm, whereas the buffer in Fig. 1(a) was marked as AlGaN layer. We suggested that the authors should check the material of buffer layer.

3.      In Page 3, line 148, the authors stated that “The selection of the gate metal and gate field plate structures should be carefully…” In this study, the Ni/Au is utilized as the gate field plate material, and a large-area structure with a width of 11 µm was designed. Nevertheless, the reasons were not further analyzed, and what about the other gate metal? Could the authors provide more details about the design process?

4.      In Page 4, line 164, the authors stated that “precise control of the AlGaN layer etch depth is critical…”, whereas the details about the depth was not provided. Could the authors further explain how the etch depth in this study is determined.

5.      In Page 5, line 215 and 219, the authors stated that “an ordinary VTH hysteresis of 0.6 V was investigated…”, “If MIS-HEMT with recessed gate device has low VTH hysteresis, further optimization…was required,” It suggested that the VTH hysteresis of this device was much better than other devices, or this device still required further optimization?

6.      In Page 6, line 235, the authors stated that “… is dominated by the source-drain leakage current, whereas the gate leakage current was well blocked by…”. In Fig. 4(b), we noticed that the gate and drain current were presented with the similar level, and it may not consistent with the above statement. In addition, we suggested that more explanations of the two-step breakdown behavior in Fig. 4(b) can be provided for better understanding of the readers.

7.      In Page 7, Table I, the parameters of this device were compared with the results in previous studies. We noticed that the Ref. [44-54] cited in the table mainly focused on the results in 2010 to 2013 and were not the recent studies. Could the authors update the literature to the results from the last five years?

In this paper, the authors demonstrate a high current density and high breakdown voltage of MIS-HEMT with recessed gate by the low damage gate recessed etching of atomic layer etching (ALE) technology. Though this work could be interesting, the reviewer thinks that it requires major revisions before it can be considered for publication. Detailed comments are listed as follows:

1.      We noticed that many illustrations in this paper were repeated and the statements should be checked carefully. For example, in Page 2 Line 74, the authors stated that “…and keeps low etching damage and keep low etching damage.” In Page 3 Line 122 and 123, the authors stated that “gate width (Wg) of 100 µm, … and the gate width (WG)of 100 µm.” In Page 3 Lin 140 and 147, the authors stated that “The gate field plate… affecting its breakdown voltage, gate leakage current, …the gate field plate structure affects the device’s breakdown voltage and gate leakage current.”

2.      In Page 2, line 91, the authors stated that the GaN buffer layer was 4 µm, whereas the buffer in Fig. 1(a) was marked as AlGaN layer. We suggested that the authors should check the material of buffer layer.

3.      In Page 3, line 148, the authors stated that “The selection of the gate metal and gate field plate structures should be carefully…” In this study, the Ni/Au is utilized as the gate field plate material, and a large-area structure with a width of 11 µm was designed. Nevertheless, the reasons were not further analyzed, and what about the other gate metal? Could the authors provide more details about the design process?

4.      In Page 4, line 164, the authors stated that “precise control of the AlGaN layer etch depth is critical…”, whereas the details about the depth was not provided. Could the authors further explain how the etch depth in this study is determined.

5.      In Page 5, line 215 and 219, the authors stated that “an ordinary VTH hysteresis of 0.6 V was investigated…”, “If MIS-HEMT with recessed gate device has low VTH hysteresis, further optimization…was required,” It suggested that the VTH hysteresis of this device was much better than other devices, or this device still required further optimization?

6.      In Page 6, line 235, the authors stated that “… is dominated by the source-drain leakage current, whereas the gate leakage current was well blocked by…”. In Fig. 4(b), we noticed that the gate and drain current were presented with the similar level, and it may not consistent with the above statement. In addition, we suggested that more explanations of the two-step breakdown behavior in Fig. 4(b) can be provided for better understanding of the readers.

7.      In Page 7, Table I, the parameters of this device were compared with the results in previous studies. We noticed that the Ref. [44-54] cited in the table mainly focused on the results in 2010 to 2013 and were not the recent studies. Could the authors update the literature to the results from the last five years?

Author Response

Dear Reviewers,

Thank you for your feedback and comments.

Reviewer 1

Comments and Suggestions for Authors

  1. We noticed that many illustrations in this paper were repeated and the statements should be checked carefully. For example, in Page 2 Line 74, the authors stated that “…and keeps low etching damage and keep low etching damage.” In Page 3 Line 122 and 123, the authors stated that “gate width (Wg) of 100 µm, … and the gate width (WG)of 100 µm.” In Page 3 Lin 140 and 147, the authors stated that “The gate field plate… affecting its breakdown voltage, gate leakage current, …the gate field plate structure affects the device’s breakdown voltage and gate leakage current.”

AnswerWe have removed the repeated sentences in line 75, 122 and 144-147.

  1. In Page 2, line 91, the authors stated that the GaN buffer layer was 4 µm, whereas the buffer in Fig. 1(a) was marked as AlGaN layer. We suggested that the authors should check the material of buffer layer.

AnswerWe have corrected the statement in the text on line 91 to the AlGaN buffer.

  1. In Page 3, line 148, the authors stated that “The selection of the gate metal and gate field plate structures should be carefully…” In this study, the Ni/Au is utilized as the gate field plate material, and a large-area structure with a width of 11 µm was designed. Nevertheless, the reasons were not further analyzed, and what about the other gate metal? Could the authors provide more details about the design process?

Answer:We have provided the reasons for selecting the gate metal and gate-field plate design on line 146-167.

The common choices of metal for the gate field plate are Ni/Au [41] or Ti/Au [42], both of which have excellent conductivity and adhesion (152-153). The gate field plate with a width of 11 μm and the field plate distance extends from the edge of the gate to the drain (LGFP) of 3.5 μm (114-115). As indicated in the reference, the lower the proportion of the LGFP length to the LGD length, the higher the maximum transconductance (Gm) value and the gain, which is mainly due to the reduction in the access resistance caused by the increase in available carriers in the channel. According to the reference by A. Mohanbabu et.al, when the LGFP/LGD ratio is around 30%, it can achieve better Gm and breakdown voltage. [41-43]. (162-167)  

  1. In Page 4, line 164, the authors stated that “precise control of the AlGaN layer etch depth is critical…”, whereas the details about the depth was not provided. Could the authors further explain how the etch depth in this study is determined.

Answer:We employed Oxford Instruments Plasma Technology's state-of-the-art ALE equipment in this study. Oxford Instruments Plasma Technology, in collaboration with LayTec, offers critical capabilities for precisely controlling the accuracy of etched AlGaN thickness, achieving an impressive tolerance of (+/-0.5 nm) through in-situ metrology. Further details on this technology can be found in the following announcement: https://plasma.oxinst.com/media-centre/webinars/manufacturing-of-reliable-normally-off-recessed-gate-algan-gan-mishemts 

We have added how the etch depth in this study is determined for Page 4, line 182-194.

To avoid potential dielectric (Al2O3) side effects, we investigated three different depths of 5 nm, 3 nm, and over-etching of AlGaN gate recess in Schottky HEMT devices, and the reference device is without gate recessed etching. By examining the correlation between the threshold voltage shift and AlGaN depths, we found that AlGaN with 3 nm remaining thickness of the AlGaN layer shows the realization of normally-off characteristics in the HEMT devices. Although the over-etching of the AlGaN gate recess also has normally-off characteristics, the lowest drain current of 6 mA/mm. AlGaN with 3 nm remaining shows a threshold voltage of +0.5 V and drain current of 440 mA/mm at a gate voltage of 4V, and the reference device without gate recessed etching shows a threshold voltage of -4.5 V and drain current of 650 mA/mm. It's essential to find that determination to utilize the 3 nm AlGaN remaining thickness was influenced by a prior study, which yielded promising outcomes with this particular approach in recessed gate MISHEMT.

  1. In Page 5, line 215 and 219, the authors stated that “an ordinary VTHhysteresis of 0.6 V was investigated…”, “If MIS-HEMT with recessed gate device has low VTH hysteresis, further optimization…was required,” It suggested that the VTH hysteresis of this device was much better than other devices, or this device still required further optimization?

Answer:The VTH hysteresis of this device was better than other devices, and we added the description on line 263-266.

A VTH hysteresis of 0.6 V was found by high-based (up-sweep) and low-based (down-sweep). VTH hysteresis phenomenon was produced by electrons caught in the AlGaN/Al2O3 dielectric interface states or traps that remain in the Al2O3 dielectric bulk layer. Compared to other devices, the hysteresis of this device was relatively low [49, 50].

  1. In Page 6, line 235, the authors stated that “… is dominated by the source-drain leakage current, whereas the gate leakage current was well blocked by…”. In Fig. 4(b), we noticed that the gate and drain current were presented with the similar level, and it may not consistent with the above statement. In addition, we suggested that more explanations of the two-step breakdown behavior in Fig. 4(b) can be provided for better understanding of the readers.

Answer:we added the description on line 280-303.

The device's hard breakdown is dominated by the gate-to-drain leakage current (VD)=720 V at 1 μA/mm. One Al2O3 gate dielectric design layer and low surface damage recessed gate by the ALE process improved the gate leakage current. Compared to the Hsieh et al. study, using Al2O3/AlN stack insulator reduces interface trapping density between the Al2O3/GaN interface for gate-recessed GaN MIS-HEMT by low power plasma etching made, which hard breakdown occurred at gate-to-drain voltage (VD)=620 V at 2 μA/mm. High gate-to-drain leakage current mechanism, due either to surface conduction or to the tunneling of electrons through the AlGaN barrier [51]. According to gate-to-drain breakdown voltage results, there are three-step breakdown behavior. The slight decrease of IG current before VD of 200 V in the initial stages indicates trapping, and the start increase of IG current from 200 V to 1100 V after that can be attributed to stress-induced leakage current, SILC [52]. When applied to the device, electrical stress can cause structural changes in surface conduction or the oxide layer, creating traps that can capture and release charge carriers. As a result, these traps can influence the current flow through the device, leading to leakage currents. Dielectric hard breakdown, the catastrophic breakdown, is observed to take place after 1100 V and is characterized by a very sharp vertical jump in the gate current of each device [53]. Therefore, BV was exhibited in 720 V at 1 μA/mm and another BV was exhibited in 1190 V at 1 mA/mm. Owing to the low-damaged surface of the recessed gate can be well covered by the high-quality dielectric Al2O3, the high critical breakdown field strength between the gate and the drain region can be effectively suppressed by gate field plate design. Low surface gate leakage current is demonstrated to achieve the high breakdown voltage performance of 1190 V. Additionally, gate field plate design also helps achieve higher BV performance for recessed gate devices

  1. In Page 7, Table I, the parameters of this device were compared with the results in previous studies. We noticed that the Ref. [44-54] cited in the table mainly focused on the results in 2010 to 2013 and were not the recent studies. Could the authors update the literature to the results from the last five years?

Answer:We have added the results from literatures [54-62] over the past five years.

Reviewer 2 Report

After review this work, I can say that the topic is relevant for the community. However, I have some observations.

1. Since normally of HEMT it is need to add references about p-GaN gate HEMT.

2. The structure illustrated in figure 1a showed an AlGaN buffer layer. However, is more common employ AlN as a buffer layer. I suggest to add at least 2 recent references such as the following.

2022 Mater. Res. Express 9 065903DOI 10.1088/2053-1591/ac7512

3. From figure 3 is observed that the AlGaN layer is 3.7nm. However this layer well know as 20nm and the origin of 2DEalG has been attributed to AlGaN layer. So, it is important to compare the same structure with a more thick AlGaN layer.

4. The 2DEG profile by CV measurement is need to discuss deeply.

5. Equation 1 most be supported by more references and a best description of the physics must bee added.

6. I suggest to remove figure 5.

7. In the table 1, reference [51] showed better performance than this work. So, explain the differences and the benefit of your work, supported by physics and deep discussion.

8. Conclusions need to be re-written

minor revisions are needed.

Author Response

Dear Reviewers,

Thank you for your feedback and comments.

Reviewer 2

Comments and Suggestions for Authors

  1. Since normally of HEMT it is need to add references about p-GaN gate HEMT.

Answer:We have referenced and added relevant references on p-GaN, such as references [6-8]、[35-36].

  1. The structure illustrated in figure 1a showed an AlGaN buffer layer. However, is more common employ AlN as a buffer layer. I suggest to add at least 2 recent references such as the following.

AnswerWe have added references [35, 36] regarding the AlGaN buffer layer on line 91.

  1. From figure 3 is observed that the AlGaN layer is 3.7nm. However this layer well know as 20nm and the origin of 2DEG has been attributed to AlGaN layer. So, it is important to compare the same structure with a more thick AlGaN layer.

        Answer:We have added how the etch depth in this study is determined for Page 4, line         182-194.

To avoid potential dielectric (Al2O3) side effects, we investigated three different depths of 5 nm, 3 nm, and over-etching of AlGaN gate recess in Schottky HEMT devices, and the reference device is without gate recessed etching. By examining the correlation between the threshold voltage shift and AlGaN depths, we found that AlGaN with 3 nm remaining thickness of the AlGaN layer shows the realization of normally-off characteristics in the HEMT devices. Although the over-etching of the AlGaN gate recess also has normally-off characteristics, the lowest drain current of 6 mA/mm. AlGaN with 3 nm remaining shows a threshold voltage of +0.5 V and drain current of 440 mA/mm at a gate voltage of 4V, and the reference device without gate recessed etching shows a threshold voltage of -4.5 V and drain current of 650 mA/mm. It's essential to find that determination to utilize the 3 nm AlGaN remaining thickness was influenced by a prior study, which yielded promising outcomes with this particular approach in recessed gate MISHEMT.

  1. The 2DEG profile by CV measurement is need to discuss deeply.

Answer:Thank you for your valuable feedback and insightful comments on our manuscript. We sincerely appreciate the time and effort you have invested in reviewing our work. Following your suggestion to include CV measurement in our study, we would like to acknowledge its significance in providing a deeper understanding of the material and surface condition properties. We regret to inform you that, currently, our batch of devices cannot be used for CV measurements due to the following reasons:

Prior Measurements: Our research group has already conducted essential measurements, including ID-VG, ID-VD, BV, and TEM analysis, which have significantly impacted the integrity of the devices. As a result, there are no remaining intact devices suitable for performing CV measurements.

Despite this limitation, we recognize the importance of CV measurement and the valuable insights it can offer. Next study, we plan to include CV measurement as a measured baseline in our future studies. We also have considered employing advanced simulation techniques (T-CAD) to model the electrical behavior of the materials studied. Once again, we sincerely appreciate your constructive feedback, and we are committed to improving the quality and impact of our research.

  1. Equation 1 most be supported by more references and a best description of the physics must be added.

Answer:We have added references [46, 47] on line 168. And description of physics on line 194-206

  1. I suggest to remove figure 5.

Answer:We have removed Figure 5.

  1. In the table 1, reference [51] showed better performance than this work. So, explain the differences and the benefit of your work, supported by physics and deep discussion.

Answer:We have added into article on line 310-321 [62].

Compared to the device discussed in reference [51], our study exhibits a slightly higher BV of 720 V at 1 μA/mm. However, their ID, max exceeds this work by 200 mA/mm. This is due to the double-channel heterostructure design, which offers high 2DEG mobility by the lower channel heterostructure. The upper channel heterostructure is to achieve a fully recessed gate by completely etching away the AlGaN layer. This fully recessed gate structure aims to obtain a normally-off operation with a positive VTH requirement. Typically, when the gate recess is fully etched away, VTH tends to be large, leading to a lower ID, max. Therefore, the design of a double-channel heterostructure necessitates carefully considering the complex epitaxy process to attain high positive VTH and ID, max. In contrast, this work with standard one-channel heterostructure and by ALE etching retains a remaining 3.7nm of AlGaN after gate recess, providing a simple process and controllable adjusting of the VTH and ID, max..

  1. Conclusions need to be re-written

Answer:We have revised the conclusion.

Our study successfully showcased the exceptional performance of a normally-off GaN MIS-HEMT device with a recessed gate. Using novel ALE technology allowed us to precisely control the recessed gate remaining at a realization of normally-off characteristics thickness of 3.7 nm, providing distinct advantages in nanoscale control and minimal surface damage. The device exhibited high-performance characteristics, including a high maximum drain current (ID,max) of 608 mA/mm, a controllable threshold voltage (VTH) of +2.0 V, and an outstanding on/off current ratio of 109. Furthermore, the device demonstrated a remarkable off-state hard breakdown voltage (BV) of 1190 V at 1 mA/mm in the gate-to-drain of IG current. The superior BV performance can be attributed to the low surface roughness (Ra) of 0.40 nm in the recessed gate region, ensuring excellent coverage of the high-quality dielectric Al2O3. Additionally, incorporating a gate field plate with a width of 11 μm in the design effectively suppressed the high critical breakdown field strength between the gate and drain regions, further enhancing device performance. Based on these studies, we believe that normally-off GaN MIS-HEMTs with recessed gates, enabled by the innovative ALE technology, hold significant potential for applications in large-area high-power devices. These devices achieve remarkable BV and ID,max values, and maintain controllable VTH and a high on/off current ratio, making them up-and-coming candidates for next-generation power electronics applications.

Round 2

Reviewer 1 Report

The authors have made detailed and convincing response to the comments. This manuscript has been improved and I suggest that this paper can be published as it is.

Reviewer 2 Report

the authors addressed almost all of my concerns (enough). So, I recommend the work for publication.

the english is ok